# Optimizing Nitrogen Supplementation: Timing Strategies to Mitigate Waterlogging Stress in Winter- and Spring-Type Canola

**DOI:** 10.3390/plants14172641

**Published:** 2025-08-25

**Authors:** Haochen Zhao, Onusha Sharmita, Abu Bakar Siddique, Sergey Shabala, Meixue Zhou, Chenchen Zhao

**Affiliations:** 1Tasmanian Institute of Agriculture, University of Tasmania, Newnham Drive, Launceston, TAS 7248, Australia; haochen.zhao@utas.edu.au (H.Z.); onusha.sharmita@utas.edu.au (O.S.); abubakar.siddique@utas.edu.au (A.B.S.); sergey.shabala@uwa.edu.au (S.S.); meixue.zhou@utas.edu.au (M.Z.); 2International Research Centre for Environmental Membrane Biology, Foshan University, Foshan 528000, China; 3School of Biological Science, University of Western Australia, Perth, WA 6009, Australia

**Keywords:** waterlogging mitigation, macronutrient accumulation, waterlogging stress, canola

## Abstract

Canola is an important rotation crop in Australia’s high-rainfall zone (HRZ), where frequent waterlogging occurs. Due to its lack of aerenchyma, canola is more vulnerable to waterlogging. Recent studies have shown that nitrogen supplementation can benefit crop growth under waterlogging stress. However, limited reports have addressed the physiological responses and macronutrient changes in either winter or spring canola with strategically timed fertilizer applications. This study investigated the physiological and elemental responses of two canola genotypes to nitrogen application at different timings during waterlogging stress. By imposing waterlogging stress in pot-based trials for 21 days using spring-type (Dynatron) and winter-type (Nizza) canola, our results demonstrated that nitrogen application one week prior to the onset of waterlogging marginally improved soil plant analysis development (SPAD) values in the two types of canola, but only significantly enhanced stomatal conductance along with reduced photosynthetic efficiency in Dynatron at the end of waterlogging, indicating genotypic differences. Furthermore, applying fertilizer a week before waterlogging enhanced macronutrient accumulation in Dynatron, including phosphorus, potassium, magnesium, and calcium. In contrast, Nizza only exhibited a significant increase in magnesium accumulation. Fertilizer application had no effect on reducing Mn and Fe accumulation in canola, indicating that Mn and Fe toxicity, typically associated with soil waterlogging, was not a contributing factor in nitrogen-induced waterlogging alleviation. Collectively, our results demonstrated genotype-specific nutrient dynamics, which should be considered in nitrogen-induced waterlogging alleviation.

## 1. Introduction

Waterlogging is a common abiotic stress affecting the yield of canola (*Brassica napus*) in Australia’s high-rainfall zone (HRZ), especially in southern Australia [1], where the annual rainfall is between 450 mm and 900 mm [2,3]. Soils in these regions are commonly duplex soils [4,5] with sand or loamy sand on top of clay layers [5], making them prone to waterlogging [5]. As one of the important rotation crops [6,7], canola is widely grown in the HRZ in Australia. Due to the natural lack of aerenchyma, which is critical for waterlogging tolerance [8], canola is more vulnerable to waterlogging stress [9,10,11]. Under waterlogging, the yield of crops in the HRZ is majorly affected by the limitation of accessible nutrients [1,12,13]. Nitrogen is the most demanding nutrient for most canola production systems in Australia [12]. Waterlogging exacerbates nitrogen deficiency by promoting denitrification, enhancing nitrate leaching, and influencing soil microbial activity, which alters soil microorganisms, inhibits nitrification, and decreases soil nitrogen [14]. In addition to nitrogen-related challenges, waterlogging can also lead to micronutrient imbalances, particularly manganese (Mn) and iron (Fe) toxicity. Under anaerobic conditions, the solubility of Mn and Fe increases, resulting in excessive uptake by canola plants [15]. This can disrupt cellular metabolism, impair photosynthesis, and contribute to oxidative stress, further exacerbating the negative effects of waterlogging on plant growth and yield [16,17]. Plants’ symptoms of Mn toxicity include chlorosis, leaf cupping, and stunted growth, especially in acidic soils where Mn availability is elevated [16]. Similarly, Fe toxicity can impair physiological activities and plant growth, resulting in root damage [18]. Despite these known impacts, the interaction between nitrogen application timings and micronutrient dynamics under waterlogging remains underexplored. Studies have reported differences between winter-type and spring-type canola regarding their tolerance to environmental stresses [19]. These differences are potentially related to variations in root development, with winter-type canola developing more root branches and better root architecture, enhancing its ability to tolerate waterlogging [20]. However, the differential adaptations between winter and spring-type canola in response to fertilizer treatment remain unknown. Waterlogging stress decreases the SPAD (soil plant analysis development value), which reflects the leaf chlorophyll content, restrains net photosynthesis [10], and inhibits root activity due to altered respiration processes [11]. Waterlogging also negatively impacts plant nutrient uptake [11]; however, nutrient supplementation can mitigate this effect and support plant growth under waterlogging stress [21,22]. Urea application before the flowering stage can help canola recover better from waterlogging [23], as rapid nitrogen uptake mostly occurs during stem elongation [12]. Recently, increasing attention has been paid to the effectiveness of nutrient application timing on alleviating waterlogging damage in cereals [21,24,25,26]. It has been shown that increasing fertilizer rates during waterlogging up to 90 kg ha^−1^ is more beneficial than applying nitrogen post-waterlogging [21]. In canola, nitrogen fertilizer supplementation after waterlogging significantly increased plant biomass and antioxidant enzyme activity but reduced the MDA (malondialdehyde) content in leaves and roots [26], and foliar application of urea, calcium nitrate, potassium nitrate, sulfur, and tricyclazole is beneficial for canola physiological adaptations under waterlogging [24,25]. A study conducted in the HRZ (Hamilton, VIC) proved that a raised bed system with nitrogen supplementation can significantly increase canola yield [23]. However, there are still limited reports on the effects of nitrogen application in a timing-strategic manner, not to mention a thorough analysis of related macronutrient changes due to nitrogen application in canola.

To fill this research gap, we investigated the effectiveness of fertilizer applications in a timely manner on alleviating waterlogging damage in winter- and spring-type canola, aiming to understand the physiological and elemental adaptations induced that underpin nitrogen application.

## 2. Materials and Methods

### 2.1. Plant Materials, Waterlogging, and Fertilizer Treatments

Two commercial canola cultivars were used in this study, Nizza (winter type) (RAGT Australia, 13 Future Ct, Shepparton VIC 3630) and Dynatron (spring type) (RAGT Australia, 13 Future Ct, Shepparton VIC 3630). The two cultivars were selected to represent the two predominant growth habits of canola cultivated in Australia. Both cultivars are commonly grown in canola-producing regions of Tasmania and are representative of commercial varieties used in local farming systems. Although neither cultivar is known to possess specific traits conferring tolerance to waterlogging, spring-type canola is generally considered more susceptible to waterlogging and heavy rainfall due to its shallower root architecture and shorter growth cycle. In contrast, winter-type canola, with its slower growth and extended vegetative phase, tends to exhibit greater resilience under waterlogged conditions. However, no significant differences in waterlogging response have been observed between the two cultivars used in this study. The canola seeds were sterilized with 70% ethanol, rinsed thoroughly with distilled water, and pregerminated before being transplanted to square pots (20 cm diameter × 19 cm height, with a total volume of approximately 6.0 L). These pots were pre-filled with soil collected from Mt Pleasant field trials, Tasmania, AU (41.44° S, 147.14° E). The soil pH was 6.3, containing 50 mg/kg N; 89 mg/Kg P, and 67 mg/Kg K (Appendix A). The whole experiment was conducted from May 2024 to August 2024 in a net shed area with plants exposed to natural sunlight conditions. The average temperature during the experimental period was 8.3 °C, with a maximum temperature of 18.5 °C and a minimum temperature of −4 °C. For the light intensity, the average light intensity was 7.2 MJ/m^2^, with the maximum light intensity being 10.8 MJ/m^2^ and the minimum light intensity being 2.6 MJ/m^2^ (Appendix A). A base fertilizer (4 g per pot) of Elixir SUPREME 12:12:17 SOP (ElixirZorka, Serbia, https://www.elixirzorka.rs/en/products/elixir-supreme/elixir-supreme-12-12-17/, accessed on 11 July 2025), containing 12% N, 12% P_2_O_5_, 17% K_2_O, 2% MgO, 14% SO_3_, 0.02% B, and 0.01% Zn, was initially applied to each individual pot. The onset of waterlogging started when the canola reached around the four-leaf stage by referring to the waterlogging pattern in local agronomic conditions in Tasmania, Australia, where canola is usually sown from late April to early May. In wetter years, these seedlings often encounter heavy rainfall and potential waterlogging by June, when they are nearly/commonly at the four-leaf stage (https://groundcover.grdc.com.au/innovation/climate/crop-growth-stage-the-key-to-effects-of-waterlogging, accessed on 11 July 2025). Flooding stress tends to occur when canola plants reach the four-leaf stage (after three to four weeks of sowing). For the waterlogging treatment, the water level was increased in half of the pots until reaching the soil surface, creating a surface-flooding condition that was maintained throughout the 21-day treatment period, whilst the other pots were continuously maintained with bottom irrigation as the control. Although the water did not submerge the shoot tissues, this setup induced root-zone waterlogging, consistent with the definition by Soil Quality Australia, which states that waterlogging occurs when roots cannot respire due to excess water in the soil profile, even if water does not appear on the surface [27]. The total duration of simulated waterlogging was 21 days (from day 7 to day 28, as shown in the light-blue shadow in Figure 1, Figure 2 and Figure 3 combined with three fertilizer application timings: (1) 1 week before waterlogging (BN) on day 0; (2) in the middle of waterlogging, 7 days after waterlogging (MN) on day 14; and (3) no additional fertilizer application (Base) (Table 1). For the fertilizer treatment of BN and MN, 0.6 g of urea (46% N) (~90 kg/ha) was applied to each pot when needed [21]. In Tasmania’s HRZ, waterlogging events usually begin in June and may extend into July, depending on seasonal rainfall. Field observations suggest that waterlogging commonly persists for three to six weeks, particularly during wet seasons. Therefore, a 21-day duration was selected to simulate a realistic and agronomically relevant waterlogging scenario. It is also of note that the application rate of 0.6 g of urea per pot was based on a field-equivalent nitrogen application rate of 90 kg N ha^−^^1^, which has been shown in previous studies to be beneficial under waterlogging conditions [21]. This equates to approximately 0.0009 g N/cm^2^. Given that the surface area of each pot was 0.04 m^2^, and accounting for the 46% nitrogen content in urea, we calculated that 0.6 g of urea would provide the equivalent amount of nitrogen per pot. The experimental design followed 2 genotypes × 2 treatments (waterlogging and control) × 3 fertilizer timings × 4 biological replicates (4 individual pots as a biological replicate). Control plants were maintained with bottom irrigation, and only the base fertilizer was applied. Plants were selected based on visual assessment, ensuring the absence of observable stress symptoms, disease indicators, or physical damage. In order to investigate canola growth performance under waterlogging stress with fertilizer applied before waterlogging started or in the middle of waterlogging, three critical sampling points were followed: (1) at the onset of waterlogging (day 7); (2) two days post the middle of waterlogging (day 16); and (3) at the end of waterlogging (day 28).

### 2.2. Stomatal Conductance, SPAD, and Photochemical Quantum Yield of Photosystem II

To monitor crop growth under waterlogging, we conducted measurements on photosynthesis-related parameters, including stomatal conductance and the photochemical quantum yield of photosystem II (PhiPS2). A portable Licor-600 Porometer/Fluorometer (LI-COR Environmental, Lincoln, NE, USA) in auto mode was used to measure stomatal conductance and PhilPS2 in the fully expanded second-bottom leaves of canola plants. Moreover, leaf greenness was also measured in the same leaves using a SPAD meter (SPAD-502 Plus, Konica Minolta, Thermo Fisher Scientific, 5 Caribbean Drive, Scoresby, Australia) [28]. Stomatal conductance, PhilPS2, and SPAD values were measured from 9 am to 2 pm on sunny days to avoid weather variations. During the waterlogging durations, three data points were collected from both control and waterlogged plants. Measurements were conducted on day 7 (where waterlogging stress initially started), day 16 (two days after fertilizer application added on day 14 as MN), and day 28 (for the base). For each photosynthesis-related parameter, a total of 16 plants were analyzed, and 4 plants were used for elemental analysis. Relative values were calculated based on averages ± SEs using Equation (1).(1)$$\mathrm{Relative value}=\frac{\mathrm{Sample value}}{\mathrm{Mean value of control plants with only base fertiliser}}$$

### 2.3. Shoot Element Analysis

For shoot element analysis, including measuring Ca, Fe, K, Mg, Mn, Na, P, and Zn contents, two bottom leaves of canola (the 3rd or 4th fully developed leaves from the top) were sampled and analyzed using the service of Inductively Coupled Plasma Atomic Emission Spectrometry (ICP-AES) in New Town, TAS, Australia (NATA Accreditation No. 5589, Hobart, TAS, Australia 7250; https://nata.com.au/accredited-organisation/new-town-laboratory-5589-3102/, accessed on 11 July 2025). Leaf samples from both control and waterlogged canola were collected at two time points: day 16 and day 28. The dry biomass of the above-ground part of the plant was measured on days 16 and 28, and the shoot element content per plant was calculated using Equation (2).(2)Shoot element content per plant (mg)=dry biomass (g)×leaf element content (mg/g)

### 2.4. Statistics Analysis

Statistical analysis was performed by using R [29] with RStudio (Version ID: 2024.09.1-394). One-way ANOVA and pairwise *t*-tests were performed on plant physiological parameters. Before conducting ANOVA, the data were tested for heteroskedasticity and normal distribution. For data that did not meet the prerequisites for ANOVA, the Kruskal–Wallis rank-sum test was performed, followed by pairwise comparisons using the Wilcoxon rank-sum test with continuity correction. A *t*-test was performed on the SPAD value, stomatal conductance, the photochemical quantum yield of photosystem II and the contents of leaf elements. For all the analyses, standard error bars are presented. Different letters on each bar represent a significant difference with a *p*-value of less than 0.05.

## 3. Results

### 3.1. SPAD Value, Stomatal Conductance, Photochemical Quantum Yield of Photosystem II, and Biomass

Applying nitrogen fertilizer on day 0, one week before waterlogging, significantly improved SPAD values and stomatal conductance in all investigated plants, with a 5% increase in SPAD values and a 50% increase in stomatal conductance on day 7 (Figure 1a–d; Appendix A). On day 28, the SPAD values were marginally higher in BN plants compared with those from the MN and Base treatments in both cultivars (Figure 1a,b; Appendix A). For stomatal conductance, two cultivars showed different responses to N applications. On day 7, the stomatal conductance of both cultivars from BN presented high values, while on day 16, only the stomatal conductance of Nizza from the MN treatment increased significantly. Both cultivars exhibited a sharp decrease in stomatal conductance from day 16, regardless of the timings of fertilizer application. However, Dynatron from BN continued to show statistically higher stomatal conductance than base or MN plants (Figure 1c,d; Appendix A).

The photochemical quantum yield of photosystem II (PhilPS2) of the two cultivars showed different response patterns to N application timings (Figure 1e,f; Appendix A). With no extra N application (Base), the PhilPS2 of the spring-type Dynatron was not significantly affected by waterlogging, while the winter-type cultivar Nizza showed a 50% decrease on day 28 (21 days under waterlogging treatment). When N was applied one week before waterlogging (day 0) (BN), the PhilPS2 of Dynatron exhibited a significant increase on day 7 but declined sharply on day 28. On day 28, the PhilPS2 of Dynatron from the BN treatment was only about 60%, which was much lower than that of the Base treatment (~90%). The application of nitrogen in the middle of the waterlogging treatment on day 16 showed no significant impact on Dynatron’s PhilPS2 (Figure 1e and Appendix A). In contrast, N application on day 0 and day 14 showed much less impact on the PhilPS2 of Nizza, with only a slight increase observed on day 7. When N was applied on day 14, the PhilPS2 showed a slight increase on day 16. However, Nizza plants from the BN treatment exhibited marginally higher PhilPS2 levels compared with either Base or MN plants throughout the waterlogging period (Figure 1f and Appendix A). The fertilizer treatment in both cultivars was more significantly correlated with photosynthesis-related parameters in the early stage of waterlogging (at 1 week) compared with longer waterlogging (at 21 days), where the correlation effects were reduced (Appendix A). The dry biomass of Dynatron with N application on day 16 was higher than that of the others, while different N treatments induced similar dry biomass accumulation in Nizza (Figure 2).

### 3.2. Shoot Nutrient Contents Under Waterlogging

The macronutrient accumulation of phosphorus (P), potassium (K), and magnesium (Mg) in canola shoots from BN, measured on day 16 and day 28, increased compared with the Base treatment (Figure 3a,c,e,g; Appendix A). On day 16 and day 28, the K and Mg in Dynatron shoots from the BN treatment were approximately 1.6 times and 2 times higher, respectively, compared with the Base treatment (Figure 3c,e). The Ca enrichment in Dynatron shoots from the BN treatment was significantly higher than in shoots from the other fertilizer treatments, being four times greater than in shoots from the Base treatment on both day 16 and day 28 (Figure 3g; Appendix A). In contrast, Nizza from the BN treatment exhibited an increase in macronutrient accumulation only for K and Mg (Figure 3d,f; Appendix A). On day 16, the accumulation of K, Mg, and Ca in Nizza from BN was slightly higher, being around 1.3 times greater than that from the Base treatment (Figure 3d,f,h). The amount of P accumulated in Nizza from the BN treatment was slightly lower on day 16 and about half on day 28 compared with the Base treatment (Figure 3b). Similar to Dynatron from MN, the macronutrient accumulation in Nizza from MN was lower than that in Nizza from Base, except for P, which had a similar amount to that in Nizza from Base.

The micronutrient accumulation of manganese (Mn) in Dynatron shoots was lower in BN compared with Base on day 16, while iron (Fe) and zinc (Zn) showed similar accumulation levels (Figure 4a,c,e; Appendix A). However, Fe, Mn, and Zn were enriched in BN compared with plants from the other fertilizer treatments after waterlogging on day 28, with Fe being significantly higher (Figure 4a,c,e). The micronutrient accumulation in Dynatron shoots from the MN treatment had a limited effect, with all micronutrient levels being lower compared with those from the Base and BN treatments (Figure 4a,c,e,g). Nizza from the BN treatment showed a higher level of Fe enrichment, approximately three times that of Nizza from the Base treatment, while Mn and Zn showed similar levels on day 16 (Figure 3b,d,f). After waterlogging on day 28, Nizza from BN still had the highest Fe level among all the fertilizer treatments (Figure 4b; Appendix A). The Mn accumulation in Nizza from BN on day 28 was slightly lower than in the Base treatment, while the Zn levels were similar to those in Nizza from MN. Similar to Dynatron shoots, Nizza from MN had lower micronutrient accumulation than that from the Base treatment for all three elements tested. Both cultivars’ sodium (Na) accumulation in BN had increased on day 16 and remained high after waterlogging on day 28. The Na content in Dynatron from the BN treatment was more than three times higher than that in Dynatron from Base, while Nizza from BN had about twice the amount on day 16. After waterlogging on day 28, the amount of Na from BN in both cultivars decreased slightly but remained significantly higher than that from the other two fertilizer treatments (Figure 4g,h; Appendix A). Interestingly, the Na levels in both cultivars from the MN treatment were about the same as those in plants from the Base treatment on day 28. Based on the correlation analysis, the nutrient content was mostly correlated with photosynthesis-related parameters measured, especially in the early stage of waterlogging (1 week) (Appendix A).

## 4. Discussion

### 4.1. Waterlogging Alleviation by Nitrogen Is Partially Linked to Photosynthetic Mechanisms and the Nature of Winter or Spring Characteristics in Canola

Studies have shown that plant leaves with nitrogen supplementation under waterlogging exhibit a higher chlorophyll content and stomatal conductance compared with those without additional nitrogen supplementation [22,30,31]. Our findings partially align with this claim, where both cultivars’ SPAD values were marginally higher in BN during the middle of waterlogging, and stomatal conductance remained relatively higher under the BN treatment compared with the other two fertilizer timings throughout the 21 days of waterlogging in Dynatron (Figure 1c). However, after 21 days of waterlogging, the SPAD values and stomatal conductance of both genotypes declined, showing the limitations of fertilizer timings on crop growth. At the end of waterlogging, the stomatal conductance of both cultivars dropped significantly to less than half compared with plants under control conditions (Figure 1c,d), which is a common symptom potentially related to reduced root hydraulic conductivity and transport/production of abscisic acid (ABA) [32,33]. The PhiPS2 results indicate that after 9 days of waterlogging (on day 16), different fertilizer application timings showed less difference (Figure 1e,f), and unsurprisingly, at the end of the waterlogging on day 28, plants under different fertilizer treatments all indicated reductions in the PhilPS2. This is consistent with previous studies, where plant PhilPS2 was highly inhibited if long-term waterlogging stress existed [34]. Furthermore, while both genotypes showed an immediate positive response to the nitrogen supplement (Figure 1e,f) before waterlogging, similar to a study that demonstrated an increase in PhilPS2 when nitrogen supplementation was applied in wheat [35], differential adaptations to nitrogen fertilizer application were observed between the two genotypes. In the middle of waterlogging on day 16, Dynatron, with nitrogen fertilizer applied before waterlogging, had significantly more biomass accumulation compared with plants with only the base fertilizer (Appendix A). This was potentially due to canola plants with nitrogen supplementation maintaining a higher chlorophyll content in the leaves, having enhanced photosynthesis as well as enhanced antioxidant enzyme activity, such as SOD, CAT, and POD [26]. The combined effect led to plants obtaining more energy and reduced oxidative stress, which contributed to greater biomass accumulation in Dynatron. Meanwhile, the supplement of nitrogen fertilizer 1 week before the implementation of waterlogging had only a limited effect on Nizza (Appendix A). This can be explained by the different growth patterns between winter- and spring-type plants. Dynatron is a spring-type canola, having quicker biomass acclamation in the seedling period, while Nizza is a winter-type canola, having a lower growth rate compared with Dynatron. Collectively, our results demonstrated that nitrogen supplementation one week before waterlogging induced certain positive effects on canola’s photosynthetic rate (the SPAD value, stomatal conductance, and PhilPS2), with a differential adaptive pattern observed between winter- and spring-type canola. Although nitrogen applications before waterlogging could not completely mitigate all the negative effects of waterlogging on photosynthesis, it did induce higher photosynthetic and biomass performance than nitrogen applications in the middle of waterlogging.

### 4.2. BN Alleviates Waterlogging Stress, Highly Linked to K, Mg, Ca, and P Regulations in Plants

Potassium (K) regulates osmotic pressure, ion balance, the opening and closing of stomata, and cell elongation [36]. The K concentration in plant tissues is closely linked to photosynthesis through the regulation of stomatal aperture, thus determining CO_2_ absorption and sunlight interception during photosynthesis [37]. A study found that plant K levels are positively correlated with NO_3_^−^ abundance, while the presence of NH_4_^+^ reduces K uptake [38]. Under aerobic conditions, urea undergoes hydrolysis to form NH_4_^+^, which is then nitrified into NO_3_^−^ [39,40]. This explains why plant K levels were enhanced by nitrogen applied before waterlogging (BN) compared with the Base treatment (Figure 3c,d), even though the same amount of K was applied. Meanwhile, nitrogen applied during waterlogging (MN) is likely to undergo dissimilatory nitrate reduction to NH_4_^+^ due to the anaerobic conditions [41], which limits the availability of NO_3_^−^ for plants, consistent with the observations that a slightly lower K level was observed in MN (Figure 3c,d).

Magnesium is versatile for plants as it is involved in the activity of more than 350 enzymes and is an important component of the chlorophyll molecule [42]. Magnesium availability in soil can be low under waterlogging conditions, as Mg^2+^ is mobile and can be easily leached out, leading to potential Mg deficiency [43]. BN increased Mg accumulation in all investigated plants (Figure 3e,f). The relatively higher Mg levels in canola leaves across the BN treatment may be related to nutrient remobilization or autophagy mechanisms when canola plants face waterlogging stress [44,45]. Calcium is also closely related to plant photosynthesis, as Ca^2+^ is diuretically related to photosystem II and carbon assimilation [46]. The essential nutrient Ca in both cultivars was well maintained by the BN treatment after 9 days of waterlogging (observed on day 16) but with only Dynatron consistently presenting enhanced Ca on day 28 at the end of waterlogging, partially suggesting that BN can improve Ca accumulation in canola leaves (Figure 3g,h) but with variations among genotypes. We propose that BN-induced waterlogging alleviation involved Ca uptake in both spring- and winter-type canola in the early waterlogging stage [47]. Collectively, BN’s alleviation of waterlogging stress was potentially associated with relatively higher levels of K, Mg, and Ca compared with MN or Base plants.

Phosphorus is also an essential nutrient in plants that is highly linked to photosynthesis [48]. The Dynatron plants from BN exhibited an increased P level compared with plants under the other fertilizer treatments after 9 days of waterlogging (Figure 3a). A study found that nitrogen application can promote P uptake under P-abundant conditions [49], which explains why plants from both BN and MN had higher P levels compared with those from Base. However, the opposite trend was observed in Nizza, where plants from the MN and Base treatments had higher P levels than those from the BN treatment (Figure 3b). This was potentially due to the genotypic differences between the cultivars, where Dynatron is a spring-type canola and Nizza is a winter-type canola, and the specific mechanism remains to be further studied.

### 4.3. Nitrogen-Induced Waterlogging Alleviation in Canola Is Not Related to Toxic Element Accumulations

Elemental toxicity is a widely observed abiotic stress that plants face under saturated soil conditions [50]. Phosphorus deficiency has been shown to result in elevated iron accumulation due to nutrient antagonism between P and Fe, whereas the interaction between nitrogen and iron remains less clearly defined [51]. Under waterlogging stress, the soil redox potential is reduced, transforming Fe and Mn into soluble minerals [52]. Soil microorganisms also tend to donate electrons to iron minerals, reducing Fe^3+^ to Fe^2+^ [18,53]. The ferrous form (Fe^2+^) is commonly present in waterlogged soil directly transported by plant roots through iron transporters, natural resistance-associated macrophage proteins, or chelated by nicotianamine (NA), which is then accumulated in different tissues of the plants [51,53], causing iron toxicity [54,55]. This can be evidenced by the increase in the iron content in Nizza after 9 days of waterlogging (day 16), and after 21 days (day 28) of waterlogging in both cultivars (Figure 3a,b), which was likely due to the release of soluble Fe^2+^ in waterlogged soils [52,56]. Another potential risk associated with higher N application is that Fe tends to be distributed more from roots to leaves under high N levels. Accumulated NH_4_^+^ under waterlogging conditions further enhances this risk by increasing Fe availability, which can lead to excessive Fe accumulation [51]. The enrichment of Fe in both cultivars from BN under waterlogging conditions can be potentially attributed to a combination of changes in soil chemical properties and the increased energy available for nutrient uptake in canola plants due to the higher photosynthetic rate. Element accumulation due to reduction processes similar to those of iron, related to pH and redox potential, also applies to elements such as Mn and Al [54]. In low-pH soils, Al and Mn toxicity is a common issue that can inhibit root growth and lead to enzyme inactivation [57].

In contrast, Dynatron from BN had significantly lower Mn contents than plants from Base on day 16 (Figure 4c) but similar Mn contents to plants from Base and MN on day 28. The Mn content in Nizza was similar compared to plants from Base and MN, which contradicts previous studies [58,59,60] that regularly report Mn increases in plant tissue when under waterlogging stress. Given that BN plants presented favorable photosynthetic parameters with non-predictable Mn or Fe accumulation trends, we propose that BN-induced waterlogging alleviation is not linked to the mechanism of reducing Mn or Fe accumulations in canola plant tissues.

## 5. Conclusions

In this study, we investigated physiological adaptations, reflected by photosynthetic parameters and shoot elements accumulation, using two canola genotypes with different fertilizer application timings as the treatments under waterlogging conditions. Our results revealed that nitrogen application one week prior to waterlogging improved certain photosynthetic traits, such as SPAD values, stomatal conductance, and photosynthetic efficiency, particularly in the winter-type canola, Dynatron. While both cultivars showed some benefit from early nitrogen supplementation, the magnitude and consistency of responses varied between winter- and spring-type canola. Dynatron exhibited more pronounced improvements in macronutrient accumulation (K, Mg, and Ca), whereas Nizza showed inconsistent trends, particularly in phosphorus and calcium uptake. Importantly, nitrogen applications did not reduce elemental toxicity risk. In fact, elevated levels of Fe and Na were observed in both cultivars, suggesting that early nitrogen supplementation may exacerbate certain stress-related toxicities under waterlogged conditions, which needs further analysis. This study was limited to two genotypes due to restrictive policies on importing canola cultivars into Tasmania, and it did not assess yield outcomes, which may limit the generalizability of the findings. Future research should include a broader range of cultivars and field trials to evaluate long-term impacts on yield and validate the physiological benefits observed. Additionally, mechanistic studies are needed to further understand the interactions between nitrogen timing and micronutrient dynamics under waterlogging. Additionally, this study was conducted in Tasmania’s high-rainfall zone (HRZ), so the findings may be influenced by the specific local climate and agronomic conditions; thus, broader field trials spanning distinguished climates should be conducted to gain more understanding.

## Figures and Tables

**Figure 1 plants-14-02641-f001:**
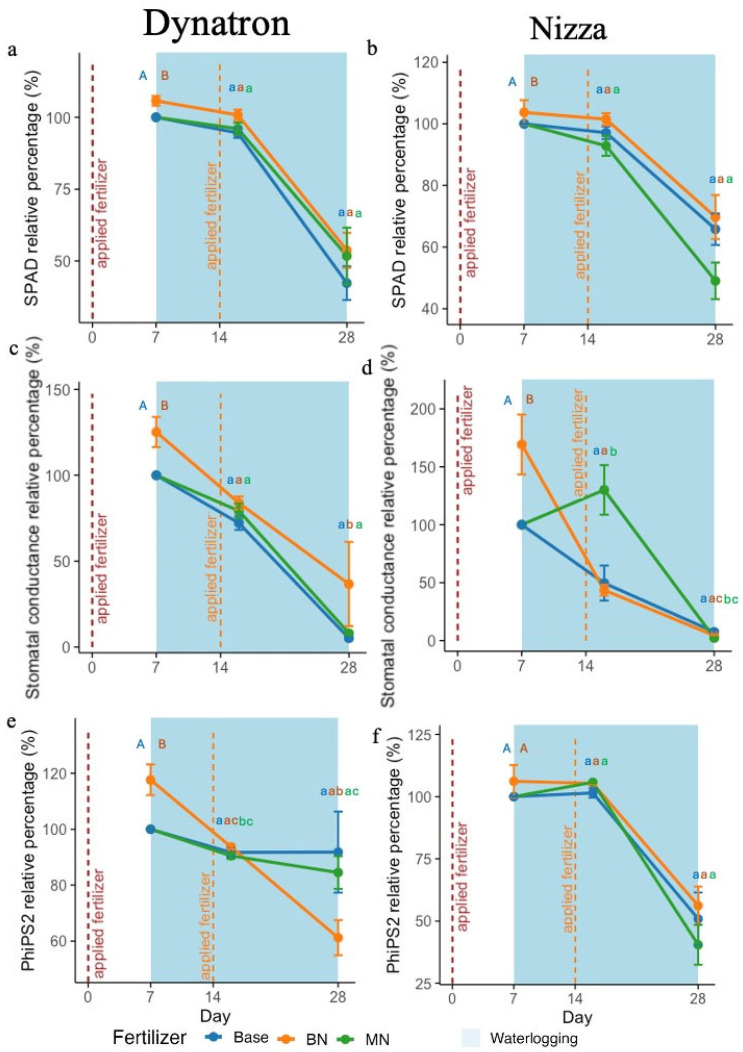
The relative percentages of SPAD (soil plant analysis development) (**a**,**b**), stomatal conductance (**c**,**d**), and PhiPS2 (photochemical quantum yield of photosystem II) (**e**,**f**) were calculated for plants under control conditions with only the base fertilizer applied. There were three fertilizer treatments: (1) Base: only base fertilizer; (2) BN: base + nitrogen fertilizer supplement before waterlogging on day 0; and (3) MN: base + nitrogen fertilizer applied on day 14 during the middle of waterlogging (7 days after waterlogging). The total duration of waterlogging stress was 14 days (from day 7 to day 28). Measurements were taken on days 0, 16, and 28. The red broken lines indicate the addition of nitrogen fertilizer on day 0 (**left**) and day 14 (**right**). Waterlogging, indicated in light blue, was implemented from day 7 to day 28. The plots are based on average values ± SEs. Different capital letters represent significant *t*-test results, while different lowercase letters indicate pairwise comparisons that are significant (*p* < 0.05).

**Figure 2 plants-14-02641-f002:**
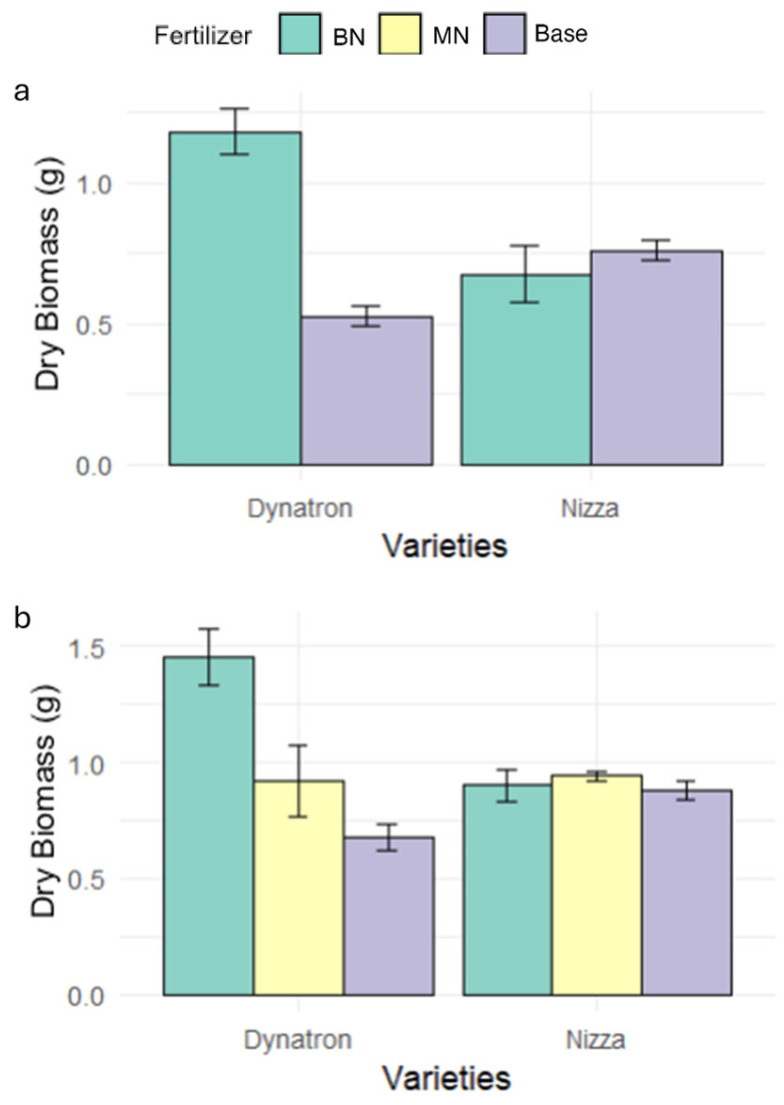
The dry biomass in the middle of waterlogging on day 16 (**a**); the dry biomass at the end of waterlogging on day 28 (**b**). There were three fertilizer treatments: (1) Base: only base fertilizer; (2) BN: base + nitrogen fertilizer supplement before waterlogging on day 0; and (3) MN: base + nitrogen fertilizer applied on day 14 during the middle of waterlogging (7 days after waterlogging). All plots are based on the average values measured with the standard errors as the error bars. Each measurement was conducted using four individual pots as biological replicates, with one plant measured per pot.

**Figure 3 plants-14-02641-f003:**
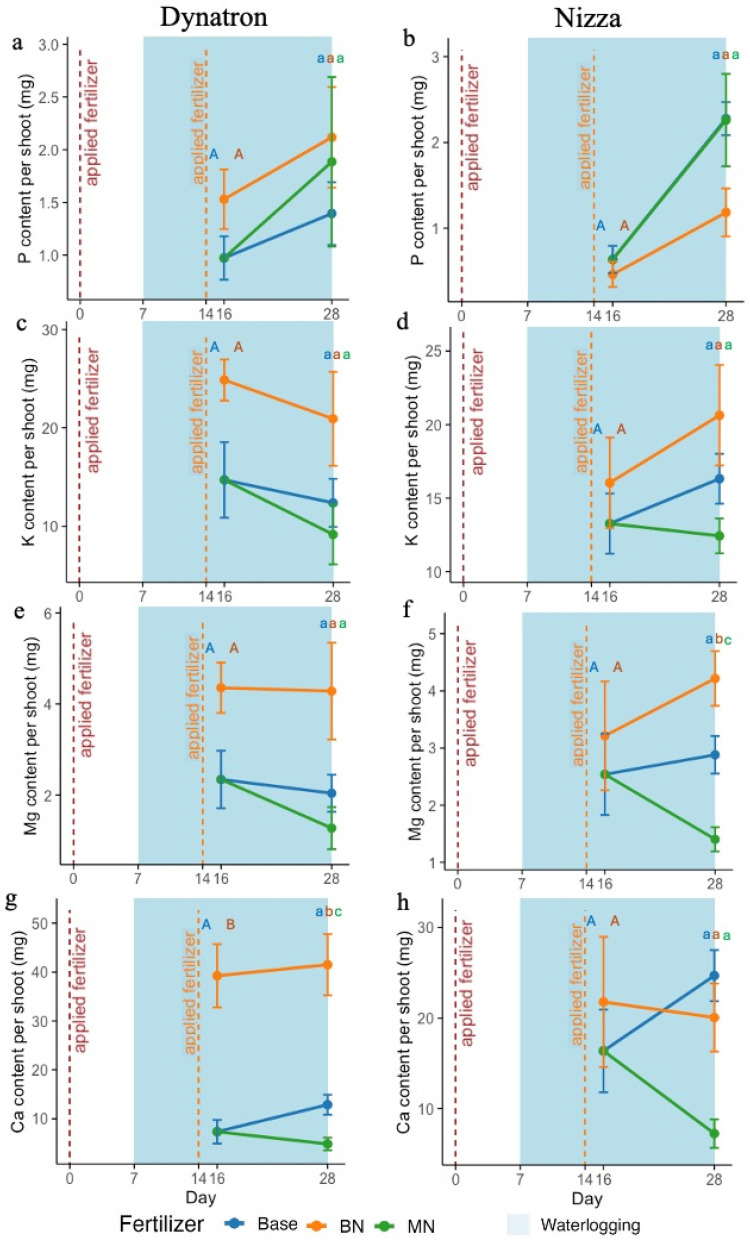
The shoot macronutrient content per plant of Dynatron and Nizza across three different fertilizer treatments: (1) Base: only base fertilizer; (2) BN: base + nitrogen fertilizer supplement before waterlogging on day 0; and (3) MN: base + nitrogen fertilizer supplement in the middle of waterlogging on day 14. Measurements were taken on days 16 and 28. (**a**,**b**) present phosphorus content per plant shoot in Dynatron and Nizza; (**c**,**d**) present potassium per plant shoot in Dynatron and Nizza; (**e**,**f**) presents magnesium per plant shoot in Dynatron and Nizza; (**g**,**h**) present calcium content per plant shoot in Dynatron and Nizza. The red lines indicate the application of nitrogen fertilizer on days 0 (**left**) and 14 (**right**). Waterlogging, indicated in light blue, was implemented from day 7 to day 28. The plots are based on average values ± SEs. Different capital letters represent significant *t*-test results, while different lowercase letters indicate pairwise comparisons that are significant (*p* < 0.05).

**Figure 4 plants-14-02641-f004:**
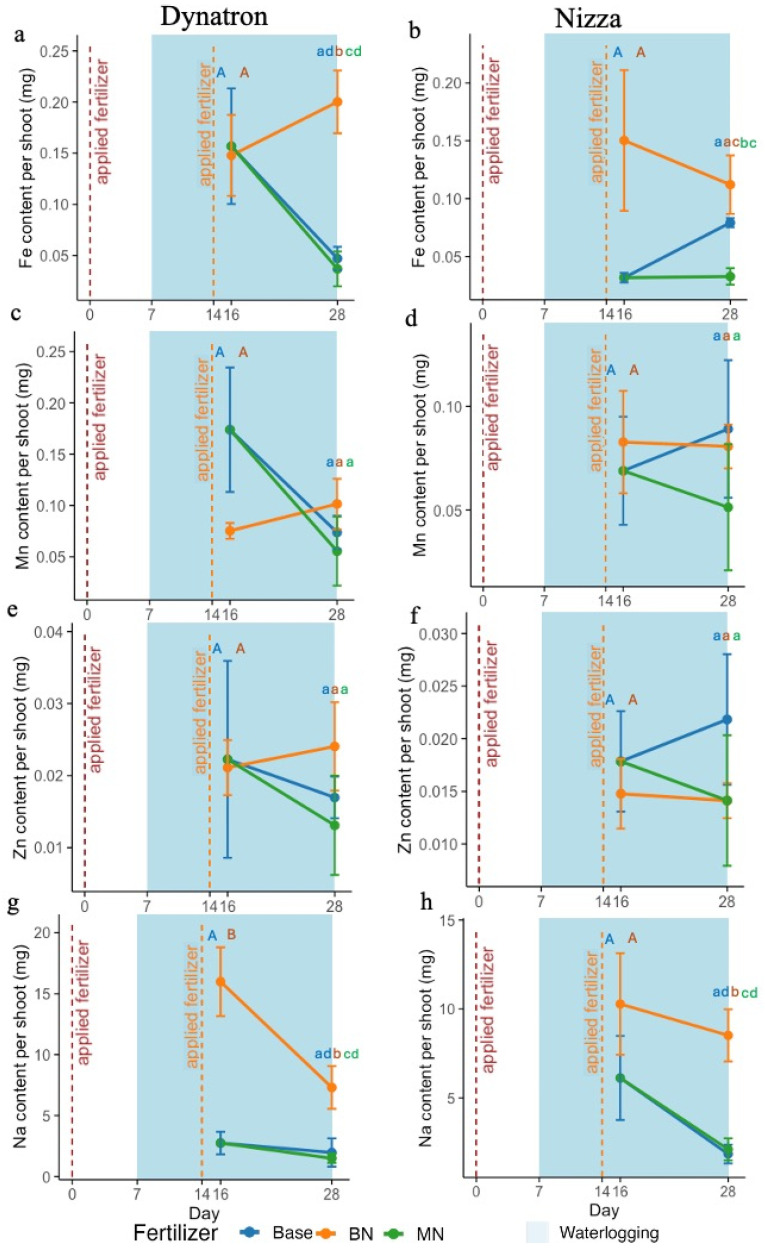
The micronutrients and sodium (Na) content per plant shoot of Dynatron and Nizza across three different fertilizer treatments: (1) Base: only base fertilizer; (2) BN: base + nitrogen fertilizer supplement before waterlogging on day 0; and (3) MN: base + nitrogen fertilizer supplement in the middle of waterlogging on day 14. Measurements were taken on days 16 and 28. (**a**,**b**) present iron (Fe) content per plant shoot; (**c**,**d**) present manganese (Mn) content per plant shoot; (**e**,**f**) present zinc (Zn) content per plant shoot; (**g**,**h**) present Na content per plant shoot. The red lines indicate the addition of nitrogen fertilizer on day 0 (**left**) and day 14 (**right**). Waterlogging, indicated in light blue, was implemented from day 7 to day 28. The plots are based on average values ± SEs. Different capital letters represent significant *t*-test results, while different lowercase letters indicate pairwise comparisons that are significant (*p* < 0.05).

**Table 1 plants-14-02641-t001:** Experimental design and fertilizer treatments.

Varieties	Waterlogging Treatment	Fertilizer Treatment
Dynatron	Waterlogging	Base fertilizer
		Base fertilizer + 0.6 g of urea 1 week before waterlogging (Day 0)
		Base fertilizer + 0.6 g of urea on day 14 during the middle of waterlogging (7 days after waterlogging)
	Control	Base fertilizer
Nizza	Waterlogging	Base fertilizer
		Base fertilizer + 0.6 g of urea 1 week before waterlogging (Day 0)
		Base fertilizer + 0.6 g of urea on day 14 during the middle of waterlogging (7 days after waterlogging)
	Control	Base fertilizer

## Data Availability

The original contributions presented in this study are included in this article. Further inquiries can be directed to the corresponding author.

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
