# Peer review of "Optimizing Nitrogen Supplementation: Timing Strategies to Mitigate Waterlogging Stress in Winter- and Spring-Type Canola"

_plants, 2025, doi:10.3390/plants14172641_

Round 1
Reviewer 1 Report (Previous Reviewer 3)
Comments and Suggestions for Authors
The authors improved the manuscript significantly. The authors provided the detailed response to the suggestions. The Materials and Methods are significantly completed and clearly represented. Statistical analysis is added to the figures in the Results.
Author Response
Reviewer #1:
Comment:
The authors improved the manuscript significantly. The authors provided the detailed response to the suggestions. The Materials and Methods are significantly completed and clearly represented. Statistical analysis is added to the figures in the Results.
Response:
We appreciate your endorsement of our manuscript. Thank you for your time and efforts in helping us improve this manuscript!
Reviewer 2 Report (Previous Reviewer 4)
Comments and Suggestions for Authors
Review Plants 3667082
The newly submitted version of the manuscript does not contain any significant changes in the experimental part, with the exception of the correlation analyses between photosynthesis-related parameters (SPAD, stomatal conductance, and PSII) and macro- and micronutrient concentrations. Explanations of the obtained results do not clarify the understanding of the possible mechanisms of the effect of the proposed scheme of nitrogen application to reduce the effects of flood stress. In addition, the analysis of only two varieties is clearly insufficient even for preliminary physiological conclusions, since the presented data sometimes contradict the conclusions of the authors. For example, by the end of the experiment only stomatal conductance and only in the Dynatron variety exceeded the control indicators, while in the Nizza variety this indicator did not differ in all three variants of treatment. Differences between varieties were also noted for PhiPS2, which in the Dynatron variety was lower when fertilizers were applied a week before flooding compared to the control. The shoot macronutrients contents per plant of Dynatron and Nizza across three different fertiliser treatments also partly contradicted to authors conclusions since the Nizza variety had phosphorus and calcium content below control values. The authors explain this effect by the genotypic differences between cultivars. All this calls into question the authors' conclusions on benefits of suggested nitrogen application timings. In addition, the proposed nitrogen application scheme significantly worsened the content of some microelements, the concentration of which is usually associated with the deteriorating impact of flood stress.
It should be recognized in accordance with the authors' opinion that “Further field experiments focusing on yield are necessary to gain a deeper understanding of how the timing of nitrogen fertiliser supplementation can mitigate waterlogging damage” (lines 372-373)
Author Response
Reviewer #2:
Comment:
The newly submitted version of the manuscript does not contain any significant changes in the experimental part, with the exception of the correlation analyses between photosynthesis-related parameters (SPAD, stomatal conductance, and PSII) and macro- and micronutrient concentrations. Explanations of the obtained results do not clarify the understanding of the possible mechanisms of the effect of the proposed scheme of nitrogen application to reduce the effects of flood stress. In addition, the analysis of only two varieties is clearly insufficient even for preliminary physiological conclusions, since the presented data sometimes contradict the conclusions of the authors. For example, by the end of the experiment only stomatal conductance and only in the Dynatron variety exceeded the control indicators, while in the Nizza variety this indicator did not differ in all three variants of treatment. Differences between varieties were also noted for PhiPS2, which in the Dynatron variety was lower when fertilizers were applied a week before flooding compared to the control. The shoot macronutrients contents per plant of Dynatron and Nizza across three different fertiliser treatments also partly contradicted to authors conclusions since the Nizza variety had phosphorus and calcium content below control values. The authors explain this effect by the genotypic differences between cultivars. All this calls into question the authors' conclusions on benefits of suggested nitrogen application timings. In addition, the proposed nitrogen application scheme significantly worsened the content of some microelements, the concentration of which is usually associated with the deteriorating impact of flood stress.
It should be recognized in accordance with the authors' opinion that “Further field experiments focusing on yield are necessary to gain a deeper understanding of how the timing of nitrogen fertiliser supplementation can mitigate waterlogging damage” (lines 372-373)
Response:
We sincerely thank reviewer for the detailed feedback. To address your concerns, we carefully re-examined our results and substantially rewrote the manuscript with conception refined. We acknowledged the differential physiological adaptations in two canola types in response to nitrogen treatment,and proposed the potential linkage with the natural character between winter and spring type plants. Accordingly, we modified the abstract, introduction, results and discussion, aiming to present our work in a more precise tone. These genotype-specific differences, along with inconsistencies in stomatal conductance and PhiPS2, have been clarified in the revised manuscript (eg. Line 193-194; line 224-225; Line 247-250). We acknowledge that nitrogen application did not alleviate micronutrient toxicity; elevated levels of Fe and Na were observed in both cultivars, suggesting that early nitrogen supplementation may exacerbate certain stress-related toxicities (Lines 417-420).
Regarding the Discussion, we added more content on the difference between winter and spring type canola (Line 323-328; line 329-335). We also added content related to potential waterlogging tolerance differences among these two genotypes (Line 89-98), aiming to form a clearer explanation of the proposed mechanisms, particularly how early nitrogen application may enhance photosynthetic performance through improvements in SPAD values, stomatal conductance, and macronutrient uptake (K, Mg, Ca). However, we acknowledge that these mechanisms remain speculative and warrant further investigation. We admit that the use of only two genotypes limits the generalizability of our findings, and this limitation has now been explicitly stated in Conclusion (Line 417-425). We also agree with the reviewer’s emphasis on the need for field-based experiments focusing on yield and have reinforced this point in the Conclusion. Additionally, we have noted that the study was conducted in Tasmania’s High Rainfall Zone (HRZ), and that the findings may be influenced by the local climate and agronomic conditions, further limiting their applicability to other canola-growing regions (lines 426-429). We are grateful for the reviewer’s insights, and we hope our revision satisfy your concerns. And again, we appreciate your efforts that have helped us improve the clarity, balance, and scientific rigour of our manuscript.
Reviewer 3 Report (Previous Reviewer 5)
Comments and Suggestions for Authors
The research Zhao et al. aims to identify strategies to mitigate crop yield damage from waterlogging and optimize fertilizer application.
The authors improved the article after the resubmission. A lot of supporting materials were added.
Why were these two commercial canola varieties used in this study, are there differences in their habit, are there differences in their response to other types of stress conditions, such as drought? Which type of anola is more susceptible to heavy rainfall and waterlogging, winter or spring?
Figure S1 - "Before N", "Mid N" need to be deciphered.
The text of the manuscript should be proofread and the style of the text should be corrected.
The Abstract and Conclusion should be revised. The Abstract does not reflect all main results and conclusions of this work. It should be revised. The Conclusion should also be expanded. Information on limitations of the work should be added.
Author Response
Reviewer #3:
Comment:
The research Zhao et al. aims to identify strategies to mitigate crop yield damage from waterlogging and optimize fertilizer application.
The authors improved the article after the resubmission. A lot of supporting materials were added.
Why were these two commercial canola varieties used in this study, are there differences in their habit, are there differences in their response to other types of stress conditions, such as drought? Which type of canola is more susceptible to heavy rainfall and waterlogging, winter or spring?
Response:
Thank you for your valuable comments. We have revised the Methods section to clarify the rationale for selecting the two commercial canola cultivars used in this study-Nizza (winter type) and Dynatron (spring type). These cultivars were chosen to represent the two predominant growth habits of canola cultivated in Australia and are commonly grown in canola producing areas of Tasmania. Although neither cultivar is known to possess specific traits conferring tolerance to waterlogging, spring-type canola is generally considered more susceptible to waterlogging and heavy rainfall due to its shallower root architecture and shorter growth cycle. In contrast, winter-type canola, with its slower growth and extended vegetative phase, tends to exhibit greater resilience under waterlogged conditions. However, no significant differences in waterlogging response were observed between the two cultivars in this study. More information can be found in lines 89-99.
Figure S1 - "Before N", "Mid N" need to be deciphered.
Response:
Thank you for pointing this out. We have revised the figure legend for Figure S1 to clarify the meaning of the fertiliser treatments used throughout the manuscript. Specifically, the legend now includes the following description: “There are three fertiliser treatments: (1) Base: only base fertiliser; (2) BN: Base + nitrogen fertiliser supplement before waterlogging at day 0; (3) MN: Base + nitrogen fertiliser applied on day 14 during the middle of waterlogging (7 days after waterlogging).” This revision aligns the figure with the rest of the manuscript and provides clearer context for interpreting the treatment labels.
The text of the manuscript should be proofread and the style of the text should be corrected.
Response:
The manuscript has now been thoroughly proofread to improve clarity and consistency in style by a few coauthors. Additionally, the text formatting has been standardized using Palatino Linotype font to ensure uniformity throughout the document. The revised content was highlighted as red font color.
The Abstract and Conclusion should be revised. The Abstract does not reflect all main results and conclusions of this work. It should be revised. The Conclusion should also be expanded. Information on limitations of the work should be added.
Response:
In response, we have substantially revised both the Abstract and Conclusion to better reflect the key findings, limitations, and implications of our study. In the revised Abstract, we now clearly summarise the main physiological and nutrient-related outcomes, including the differential responses between the two canola genotypes (Dynatron and Nizza), the observed improvements in photosynthetic traits and macronutrient accumulation under early nitrogen application, and the lack of mitigation of micronutrient toxicity (lines 15-19, 20-22, 24-26). We also highlight the need for further field-based validation and broader genotype screening (lines 420-425). The Conclusion has been expanded to provide a more balanced interpretation of the results. We now explicitly acknowledge the limitations of the study, including the use of only two genotypes, the absence of yield data, and the regional specificity of the experiment conducted in Tasmania’s High Rainfall Zone (HRZ) (lines 420-429). These limitations have been discussed in the context of their impact on the generalisability of the findings. Additionally, we have clarified that while early nitrogen application showed some physiological benefits, it also led to increased accumulation of potentially toxic elements such as Fe and Na, which warrants further investigation. We hope these revisions address the reviewer’s concerns and improve the clarity and completeness of the manuscript.
Reviewer 4 Report (Previous Reviewer 6)
Comments and Suggestions for Authors
The Authors provided reasonable and comprehensive responses to the posed initial comments and questions on their submitted Manuscript. The responses partially closed the gaps in the logical schemes and data revealed. However, the present Reviewer suggest more revisions for the Manuscript for being considered for publication.
Major points.
1) The text is difficult for understanding. It starts from the derivative parameters without providing the whole background for the story.
For example, the Reviewer who worked with colleagues in waterlogging tolerance was not aware about Mn and Fe toxicity under waterlogging while aerenchyma development and changes in respiration were among the main adaptation features.
2) Please, add more to the introduction and references about Mn and Fe toxicity for canola or similar plants under waterlogging.
3) There is an inconsistency between the experimental description and table 1.
“Control plants were 108 maintained with bottom irrigation and only base fertiliser applied. In order to investigate 109’
Table 1 show 3 treatments for control.
How could it be given?
4) Experimental design. Please, add when the dry weight of plants was measured.
5) Figures 1-3. At day 0 all the parameters were equal. Is it correct? It makes sense to add an extra point at 100% for all the parameters at day 0. Were the parameters measured before the treatment? How the plants were selected for the experiments?
6) It’s reasonable to add figure S1 and S4 to the main text before the other figures to characterize the growth of plants and elements in them which are the initial most important parameters to see and discuss.
7) “4.3 Nitrogen induced waterlogging alleviation in canola is not related to toxic element accumula- 328 tions. 329 Elemental toxicity is a common abiotic stress that plants face under saturated soil 330 conditions [46]. Studies have found that P deficiency leads to intensive Fe accumulation 331 due to the antagonistic relationship between P and Fe, while the N-Fe balance is not yet 332 fully understood [47].Under waterlogging stress, soil redox potential was reduced, trans- 333 forming Fe and Mn menials to be soluble [48]. Soil microorganisms also tend to donate 334 electrons to iron minerals, reducing Fe3+ to Fe2+ [49,50]. The ferrous form (Fe²⁺) commonly 335 present in waterlogged soil directly transported by plant roots through iron transporters, 336”
Was it described for canola and measured in the experiments described?
Minor points.
1) Figure S1. The units for the measured dry weight are not given.
2) Onset of waterlogging statred when 81 canolar reached around four leaf stage by referring to the waterlogging pattern in local 82
Started not started. Pls, check and correct many misprints found.
3) Figure 1-3. The axes x and y are too weak, it’s not clear that the points are present before the waterlogging.
4) Please, mention that the experiments are over winter time which is unusual for most areas. Please, indicate why the time was chosen when the temperatures are quite low and the growth is also low.
How is it relevant to the agricultural conditions?
5) “The average temperature during the 76 experimental period is 8.3°C, with the maximum temperature of 18.5°C and the minimum 77 temperature as -4°C.”
It’s like the study of cold tolerance. Please, discuss.
6) “For waterlogging treatment, water level was 88 increased in half of the pots until reaching soil surface, creating a surface flooding 89 condition that was maintained throughout the 21-day treatment period, whilst the other 90 pots were continuously maintained with bottom irrigation as control.”
So, it was not a clear waterlogging when the plants are covered with water but the waterlogging for roots only,
There is no need for aerenchyma is stem tissues then. Please, discuss.
7) Applying fertilizer under waterlogging in the middle of the treatment, it should be the different sort of application since roots under water, so bacterial community near roots differs.
Please, discuss.
Comments on the Quality of English LanguageReadable and well written, but could be polished.
Author Response
Reviewer #4:
Comment:
The Authors provided reasonable and comprehensive responses to the posed initial comments and questions on their submitted Manuscript. The responses partially closed the gaps in the logical schemes and data revealed. However, the present Reviewer suggest more revisions for the Manuscript for being considered for publication.
Major points.
1) The text is difficult for understanding. It starts from the derivative parameters without providing the whole background for the story.
For example, the Reviewer who worked with colleagues in waterlogging tolerance was not aware about Mn and Fe toxicity under waterlogging while aerenchyma development and changes in respiration were among the main adaptation features.
Response:
Thank you for your valuable feedback. We recognize the importance of providing a clear and accessible introduction, especially when discussing complex physiological responses. In particular, we now highlight that, beyond nitrogen related challenges, waterlogging can also lead to micronutrient imbalances specifically manganese (Mn) and iron (Fe) toxicity (lines 47-57). These aspects are often less emphasized in general discussions of waterlogging tolerance, which typically focus on morphological adaptations such as aerenchyma formation and altered respiration. By incorporating this information, we aim to offer a more comprehensive background that supports the rationale for exploring the interaction between nitrogen application timing and micronutrient dynamics under waterlogged conditions. We hope this revision improves the clarity and relevance of the introduction for readers with diverse research backgrounds.
2) Please, add more to the introduction and references about Mn and Fe toxicity for canola or similar plants under waterlogging.
Response:
Thank you for your suggestion to strengthen the introduction with additional references (lines 47-57) on Mn and Fe toxicity in canola or similar species under waterlogging conditions. In response, we have incorporated a few additional references that provide further evidence and context regarding the physiological impacts of Mn and Fe toxicity under anaerobic soil conditions. These references support the discussion of micronutrient imbalances and their relevance to waterlogging stress in canola. We believe these additions enhance the depth and relevance of the introduction, helping to better frame the rationale for our study. More details can be found from lines 47-57 in this revised version.
3) There is an inconsistency between the experimental description and table 1.
“Control plants were maintained with bottom irrigation and only base fertiliser applied. In order to investigate’
Table 1 show 3 treatments for control.
How could it be given?
Response:
Thank you for pointing out the inconsistency between the experimental description and Table 1. In the revised manuscript, we have clarified that control plants maintained with bottom irrigation and only base fertiliser applied. We have updated the Table 1 to reflect this more clearly and avoid any misunderstanding. We appreciate your careful review and helpful feedback.
4) Experimental design. Please, add when the dry weight of plants was measured.
Response:
To address this, we have now specified in Section 2.3 (lines 167-177) Shoot Element Analysis that the dry biomass of the above-ground part of the plant was measured on days 16 and 28. This addition ensures that the timing of biomass collection is clearly stated and aligns with the overall experimental timeline. We appreciate your attention to detail and hope this revision improves the transparency of our methodology.
5) Figures 1-3. At day 0 all the parameters were equal. Is it correct? It makes sense to add an extra point at 100% for all the parameters at day 0. Were the parameters measured before the treatment? How the plants were selected for the experiments?
Response:
Day 0 in our study marks the point at which plants were selected for the experiment and when urea was applied to the group designated for nitrogen treatment one week prior to waterlogging (BN). At this stage, all plants were still under normal conditions, and no physiological measurements were taken. Therefore, no data were recorded or presented for day 0 in Figures 1-3, as this time point served as the starting reference for treatment application rather than a point of assessment. Plant selection was based on visual assessment to ensure uniformity in health status. Only plants that appeared healthy and free from visible stress symptoms, disease, or physical damage were included in the experiment. This clarification has been incorporated into the revised methods section to improve transparency regarding the experimental design (142-143). We appreciate your attention to this detail and hope the revisions address your concern.
6) It’s reasonable to add figure S1 and S4 to the main text before the other figures to characterize the growth of plants and elements in them which are the initial most important parameters to see and discuss.
Response:
We agree that presenting biomass data and elemental data is important for contextualizing the physiological responses observed later in the experiment. As suggested, we have added Figure 2 as to present the biomass comparison. In our current manuscript, the shoot element content per plant which calculated based on dry biomass and leaf elemental concentrations is already presented in Figure 3 and 4.
7) “4.3 Nitrogen induced waterlogging alleviation in canola is not related to toxic element accumulations. Elemental toxicity is a common abiotic stress that plants face under saturated soil conditions [46]. Studies have found that P deficiency leads to intensive Fe accumulation due to the antagonistic relationship between P and Fe, while the N-Fe balance is not yet fully understood [47]. Under waterlogging stress, soil redox potential was reduced, trans- forming Fe and Mn menials to be soluble [48]. Soil microorganisms also tend to donate electrons to iron minerals, reducing Fe3+ to Fe2+ [49,50]. The ferrous form (Fe²⁺) commonly present in waterlogged soil directly transported by plant roots through iron transporters,”
Was it described for canola and measured in the experiments described?
Response:
It refers to general physiological and biochemical processes that occur in saturated soils, rather than mechanisms specifically measured or confirmed in canola within the scope of this experiment. Elemental toxicity, particularly involving iron (Fe) and manganese (Mn), is a widely observed abiotic stress under waterlogging. These conditions lead to a reduction in soil redox potential and pH, which in turn increases the solubility of Fe and Mn, making them more bioavailable. Microbial activity further contributes to this process by facilitating the reduction of Fe³⁺ to Fe²⁺, the ferrous form that is readily taken up by plant roots. We have elaborated that Fe²⁺ can be absorbed through specific transport mechanisms, such as iron transporters and chelation by nicotianamine, and may accumulate in various plant tissues. However, while these mechanisms are well-documented in other crop species, their specific impact on canola remains less well studied.
Minor points.
1) Figure S1. The units for the measured dry weight are not given.
Response:
The units for the measured dry weight have now been added to Figure S1. Thank you for pointing this out.
2) Onset of waterlogging statred when canolar reached around four leaf stage by referring to the waterlogging pattern in local
Started not started. Pls, check and correct many misprints found.
Response:
Thank you! This has been corrected (lines 110), and the entire manuscript has undergone a thorough proofreading process to address additional typographical and grammatical issues.
3) Figure 1-3. The axes x and y are too weak, it’s not clear that the points are present before the waterlogging.
Response:
Thank you! In response, Figures 1-3 have been revised to enhance clarity. The x- and y-axes have been emphasized for better visibility and overlapping elements have been removed to ensure that the data points are clearly distinguishable.
4) Please, mention that the experiments are over winter time which is unusual for most areas. Please, indicate why the time was chosen when the temperatures are quite low and the growth is also low.
How is it relevant to the agricultural conditions?
Response:
Thank you for your insightful comment. As noted in the manuscript, the experiment was conducted during the winter period, which aligns with the typical sowing window for canola in Tasmania (south hemisphere), Australia-late April to early May. This timing reflects local agronomic practices (https://groundcover.grdc.com.au/innovation/climate/crop-growth-stage-the-key-to-effects-of-waterlogging), where early growth stages of canola frequently coincide with increased rainfall and a high risk of waterlogging, particularly by June when the plants are commonly at the four-leaf stage. The decision to conduct the experiment during this period was therefore intentional and based on regional relevance. It allowed us to simulate realistic field conditions under which waterlogging stress typically occurs in Tasmanian cropping systems. Although winter temperatures are relatively low and growth rates are slower, this timing is agriculturally significant for understanding the impact of early season waterlogging on canola establishment and development in this region.
5) “The average temperature during the experimental period is 8.3°C, with the maximum temperature of 18.5°C and the minimum temperature as -4°C.”
It’s like the study of cold tolerance. Please, discuss.
Response:
Thank you for your thoughtful observation. While the temperature conditions during the experimental period (average 8.3°C, ranging from -4°C to 18.5°C) may appear to resemble a cold tolerance study, the primary focus of this research was on waterlogging stress during early growth stages of canola under realistic field conditions in Tasmania, Australia. As outlined in the manuscript, canola in this region is typically sown in late April to early May, and waterlogging events frequently occur in June when seedlings are at the four-leaf stage. The temperature profile during this period reflects the natural environmental conditions under which waterlogging stress commonly affects crop establishment. Therefore, the study was designed to simulate these agronomic realities rather than to investigate cold tolerance. We have clarified this distinction in the revised manuscript to emphasize the relevance of the temperature conditions to local agricultural practices and the specific focus on waterlogging stress (Lines 110-123).
6) “For waterlogging treatment, water level was increased in half of the pots until reaching soil surface, creating a surface flooding condition that was maintained throughout the 21-day treatment period, whilst the other pots were continuously maintained with bottom irrigation as control.”
So, it was not a clear waterlogging when the plants are covered with water but the waterlogging for roots only, There is no need for aerenchyma is stem tissues then. Please, discuss.
Response:
Water level in our treatment did not submerge the entire plant but was raised to the soil surface, creating a condition of root-zone waterlogging rather than full submergence. As per the definition provided by Soil Quality Australia, “Waterlogging occurs when roots cannot respire due to excess water in the soil profile. Water does not have to appear on the surface for waterlogging to be a potential problem.” (https://www.soilquality.org.au/factsheets/waterlogging) (lines 120-124). This aligns with our experimental setup, where the root zone was subjected to prolonged anaerobic conditions due to saturated soil, while the shoot remained unsubmerged. However, root-zone hypoxia can still trigger anatomical and physiological adaptations in both roots and lower stem tissues, depending on species and stress duration. Our study focused on these root-level waterlogging stress responses, and we have clarified this distinction in the revised manuscript to avoid confusion regarding the extent of flooding. We appreciate your insightful feedback and have updated the manuscript to better reflect the nature of the waterlogging treatment and its implications for plant response.
7) Applying fertilizer under waterlogging in the middle of the treatment, it should be the different sort of application since roots under water, so bacterial community near roots differs.
Please, discuss.
Response:
Thank you for your insightful comment! We agree that applying fertilizer during waterlogging may influence the rhizosphere environment, particularly due to altered microbial activity and oxygen availability. These conditions can indeed affect the form and efficiency of nitrogen uptake, as well as the structure of the bacterial community near the roots. While our current study focused on the physiological responses under a standard fertilization method, we acknowledge the importance of exploring alternative nitrogen application strategies under waterlogged conditions. As such, we will incorporate this aspect in our future research, which will investigate different forms and methods of nitrogen application-including bio-fertilizers, slow-release formulations, and foliar sprays to better understand their interactions with soil microbial communities and plant nutrient uptake under hypoxic stress. We appreciate your suggestion and have noted it as a valuable direction for the next stage of our work.
Reviewer 5 Report (Previous Reviewer 7)
Comments and Suggestions for Authors
The authors have successfully addressed the requested revisions, demonstrating a thorough and thoughtful approach to refining their manuscript. The revised version effectively incorporates the suggested improvements, enhancing the clarity, rigor, and overall quality of the research.
The responses to previous comments were comprehensive and well-integrated into the manuscript, ensuring that all concerns have been adequately resolved. The current version meets the necessary standards for publication, and I confidently recommend its acceptance in its present form.
Author Response
Reviewer #5:
Comment:
The authors have successfully addressed the requested revisions, demonstrating a thorough and thoughtful approach to refining their manuscript. The revised version effectively incorporates the suggested improvements, enhancing the clarity, rigor, and overall quality of the research.
The responses to previous comments were comprehensive and well-integrated into the manuscript, ensuring that all concerns have been adequately resolved. The current version meets the necessary standards for publication, and I confidently recommend its acceptance in its present form.
Response:
Thank you for your encouraging and supportive feedback! We are especially appreciative of your recognition of the improvements made. Your support and detailed evaluation have played a significant role in shaping the final version of this manuscript.
Round 2
Reviewer 2 Report (Previous Reviewer 4)
Comments and Suggestions for Authors
Review2 Plants 3667092
The authors have substantially revised the discussion of the results obtained and the conclusion in the submitted version of the manuscript, linking optimization of nitrogen supplementation under waterlogging stress with genotype-specific nutrient dynamics of in a winter and a spring type canola. This interpretation is more consistent with the experimental data obtained by the authors. The authors also provided a rationale for why the study was limited to two genotypes. Despite the limited nature of the results and their direct connection with the specific conditions of Tasmania, they are nevertheless of certain interest to readers. I have no further objections to its consideration for publication.
Author Response
Reviewer 2
The authors have substantially revised the discussion of the results obtained and the conclusion in the submitted version of the manuscript, linking optimization of nitrogen supplementation under waterlogging stress with genotype-specific nutrient dynamics of in a winter and a spring type canola. This interpretation is more consistent with the experimental data obtained by the authors. The authors also provided a rationale for why the study was limited to two genotypes. Despite the limited nature of the results and their direct connection with the specific conditions of Tasmania, they are nevertheless of certain interest to readers. I have no further objections to its consideration for publication.
Response:
Thank you! We appreciate your recognition of this revised manuscript and are grateful for the thoughtful comments and suggestions provided.

Reviewer 4 Report (Previous Reviewer 6)
Comments and Suggestions for Authors
The Authors reasonably responded to the posed questions and made positive amendments which improve the clarity of the Manuscript. The volume of information is sufficient for being published.
Still the present Reviewer suggests to add some revisions to improve the text.
Minor and major points.
1) The total duration of waterlogging stress were 14 days (From day 7 to day 21). Measurements were taken 207
Misprint , was 14 days, not were.
2) This study was limited to two genotypes due to the restrict policy in importing 420 canola cultivars to Tasmania, and did not assess yield outcomes, which may restrict the 421 generalisability of the findings.
Please, check language.
3) Figure 1.
The Figure became better for understanding. Did the Authors do measurements at time point 0? It would be good to add point of 0 with 100% for all the curves. However, it should not be added if the measurements were not done.
4) Base fertiliser of 4g (12%N, 12%P, 12%K, 2%MgO, 14%S, 0.02%B, and 0.01%Zn) was 109 initially added to each individual pot.
Please, indicate what was the fertilizer; the producing company, commercial name etc.
5) Figure 1. Do the Authors have exact data in absolute values, not in %? It would be good to add them if available.
6) Table S1, soil properties. Please, indicate where the soil came from. Please, add statistics for the soil measured parameters. Please, add the part in the methods.
7) Table S1. Heading, values, not “valus”.
8) PhilPS2 is better to give the full name of the parameter in figure legends and at the axes.
9) Figure 3. Figure y axis. Not per a plant but per a shoot or so. The roots were not analyzed while they could be the most important part of the plants under the treatments.
10) Figure 4. It might be helpful to use the colour scheme for indicating the fertilizers. Namely, to give the words “applied fertilizer” with lines at point 0 with yellow-orange colour, same as the line for the treatment. Then to use green colour for “the words “applied fertilizer” with lines at point 14, same as the corresponding line.
11) Abstract.
more, applying fertilizer a week before waterlogging enhanced macronutrient accumula-24 tion in Dynatron, including phosphors, potassium, magnesium and calcium. In contrast, 25
Misprint, to be phosphorus.
Please, check all the text.
12) The Reviewer would like to see much more complex analysis with many more parameters measured, stressing the parameters of plants and making an accent on plant roots. At least, for the future.
Comments on the Quality of English LanguageMisprints and simple errors are easily seen in the new written added parts.
Author Response
Reviewer 4
The Authors reasonably responded to the posed questions and made positive amendments which improve the clarity of the Manuscript. The volume of information is sufficient for being published.
Still the present Reviewer suggests to add some revisions to improve the text.
Response:
We are glad to hear your endorsement regarding the clarity and content of this revised manuscript. We also appreciate your suggestions for some further amendments. Regarding this, we carefully addressed your comments, either by applying amendments in the main context or with further explanation. We hope these revisions satisfy your concerns and further enhance the quality of this manuscript.
Minor and major points.
1) The total duration of waterlogging stress were 14 days (From day 7 to day 21). Measurements were taken 207
Misprint , was 14 days, not were.
Response:
Thank you! This has been corrected (line 210).
2) This study was limited to two genotypes due to the restrict policy in importing 420 canola cultivars to Tasmania, and did not assess yield outcomes, which may restrict the 421 generalisability of the findings.
Please, check language.
Response:
Thank you for pointing this out. We have revised the sentence as “This study was limited to two genotypes due to restrictive policies on importing canola cultivars into Tasmania, and it did not assess yield outcomes, which may limit the generalisability of the findings.” (Lines 423-425). We have also carefully checked the entire manuscript to correct similar issues and improve overall readability. We appreciate your attention to detail and thank you for your helpful feedback.
3) Figure 1.
The Figure became better for understanding. Did the Authors do measurements at time point 0? It would be good to add point of 0 with 100% for all the curves. However, it should not be added if the measurements were not done.
Response:
Thank you for your thoughtful comment regarding the baseline values and the timing of measurements in Figure 1. In our study, day 0 marks the point at which plants were selected for the experiment and when additional urea was applied to the group. At this stage, plants were not reaching a proper growth stage for treatment and thus were still under normal watering conditions. Therefore, no physiological measurements (SPAD, stomatal conductance, or ΦPSII) were taken on day 0 in Figures 1, 3and 4. We appreciate your attention to this detail and hope our explanation address your concern.
4) Base fertiliser of 4g (12%N, 12%P, 12%K, 2%MgO, 14%S, 0.02%B, and 0.01%Zn) was 109 initially added to each individual pot.
Please, indicate what was the fertilizer; the producing company, commercial name etc.
Response:
Agree. We have now updated text as: “Base fertiliser (4 g per pot) of Elixir SUPREME 12:12:17 SOP (Elixir Zorka, Serbia, https://www.elixirzorka.rs/en/products/elixir-supreme/elixir-supreme-12-12-17/), containing 12% N, 12% P₂O₅, 17% K₂O, 2% MgO, 14% SO₃, 0.02% B, and 0.01% Zn, was initially applied to each individual pot.” With the weblink provided, it ensures that readers can easily access to the fertilizer information for their convenience (Lines 109-111).
5) Figure 1. Do the Authors have exact data in absolute values, not in %? It would be good to add them if available.
Response:
We have added Table S6 (now Table S6 in the Supplementary Material) presenting the absolute values of SPAD, stomatal conductance, and ΦPSII under the three fertiliser treatments. Data are presented as mean ± SE. We hope this addition provides the clarity requested and supports the better interpretation of the results.
6) Table S1, soil properties. Please, indicate where the soil came from. Please, add statistics for the soil measured parameters. Please, add the part in the methods.
Response:
Agree. We have clarified in the Methods section (Lines 101-102). Soil used in this experiment was collected from the Mt Pleasant field trials in Tasmania, Australia (41.44° S, 147.14° E). To improve transparency, we have also updated Table S1 to include the full nutrient profile and clarified that all parameters were measured prior to the waterlogging. Soils were collected, dried, sieved, and thoroughly mixed to ensure homogeneity. Three subsamples were taken from different parts of the soil stock and combined to form a composite sample for analysis. As a result, no variation was observed across replicates, and statistical values (e.g., standard deviation) were not applicable for this dataset. We appreciate your attention to these details and have incorporated the necessary clarifications into both the table and the methods section.
7) Table S1. Heading, values, not “valus”.
Response:
Thank you! This has been corrected.
8) PhilPS2 is better to give the full name of the parameter in figure legends and at the axes.
Response:
Due to space limitations and to maintain the visual clarity of the figure, we have retained the abbreviation “PhiPS2” on the axes. However, we have updated the figure title to include the full name for clarity: “PhiPS2 (Photochemical quantum yield of photosystem II)” (Lines 206-207). We hope this revision addresses your concern while preserving the readability and aesthetics of the figure.
9) Figure 3. Figure y axis. Not per a plant but per a shoot or so. The roots were not analyzed while they could be the most important part of the plants under the treatments.
Response:
Thank you for pointing this out. We agree that the values presented are per shoot rather than per whole plant. Therefore, we have changed the y axis as ‘…per shoot’ in these revised Figure 3 and 4.
10) Figure 4. It might be helpful to use the colour scheme for indicating the fertilizers. Namely, to give the words “applied fertilizer” with lines at point 0 with yellow-orange colour, same as the line for the treatment. Then to use green colour for “the words “applied fertilizer” with lines at point 14, same as the corresponding line.
Response:
Agree. We have changed all the figures in the context according to your suggestions.
11) Abstract.
more, applying fertilizer a week before waterlogging enhanced macronutrient accumula-24 tion in Dynatron, including phosphors, potassium, magnesium and calcium. In contrast, 25
Misprint, to be phosphorus.
Please, check all the text.
Response:
Thank you! We have corrected “phosphors” to “phosphorus” in the revised sentence. We have also carefully checked the manuscript for similar errors to ensure consistency and accuracy throughout the text.
12) The Reviewer would like to see much more complex analysis with many more parameters measured, stressing the parameters of plants and making an accent on plant roots. At least, for the future.
Response:
Thank you for your valuable suggestions! We fully agree that root traits play a critical role in canola responses to waterlogging and nutrient dynamics. While the current study focused on aboveground physiological and nutrient responses, we recognise the importance of root-level processes and will incorporate detailed root measurements and additional physiological parameters in future experiments, aiming at uncovering root derived waterlogging resistance mechanisms in canola.

This manuscript is a resubmission of an earlier submission. The following is a list of the peer review reports and author responses from that submission.
Round 1
Reviewer 1 Report
Comments and Suggestions for Authors
In this manuscript, the authors present data on the effect of added nitrogen fertilizer on the physiological parameters of canola during waterlogging. The results contribute to understanding how pre-applied nitrogen influences the waterlogging tolerance of canola. However, the findings remain pretty preliminary. The data are insufficient to provide significant insights. In my opinion, merely detecting the effects of fertilization on several physiological parameters (such as SPAD, stomatal conductance, etc.) and macro- or micro-element content is inadequate for drawing meaningful conclusions. The mechanism by which nitrogen addition promotes waterlogging tolerance has not been thoroughly investigated or discussed. To my knowledge, element deficiency is not the primary cause of damage due to waterlogging stress. The potential mechanism of nitrogen-induced alteration in macro- or micro-element accumulation, as presented by the authors, remains largely speculative. To clarify the mechanism behind nitrogen-enhanced waterlogging tolerance, the authors should provide considerably more data.
Reviewer 2 Report
Comments and Suggestions for Authors
The graphs presented in the study need changes 1) add significance marks 2) no need to repeat BN,BM Base in all graphs, since all have similar color and symbols. 3) font size can be increased to 10-12 for better vizulization 4) in line plot the width of lines should be adjusted.
why bottom leaves were selected. for such experiments 3rd or 4th fully develope leaves from top should be selected. digestion mrthods and nutrient estimation should be explained in details. add instrument model number and details.
Reviewer 3 Report
Comments and Suggestions for Authors
This study represents the results of application nitrogen fertilizers for better adaptation of canola plants to the waterlogging stress. Plants of two cultivars were grown in the pots, three time points were used for application of the Nitrogen fertilizer (0.6 g urea) . To assess the effect of nitrogen application, Stomatal conductance, SPAD and photochemical quantum yield of photosystem II, cations were analyzed in the shoots and leaves. The results showed that nitrogen application one week before waterlogging significantly improves the plant adaptation to waterlogging stress. These results will be useful to further clarify the mechanisms of plant adaptation to waterlogging stress. Several studies have addressed this problem before. The novelty of the current research is evaluate the time-dependent changes of N-application and reveal its effect on macronutrient content under waterlogging.
The article needs the following revisions:
Introduction: What is the SPAD abbreviation and why this parameter is reliable – has to be described
Materials and Methods:
- The volume of the soil in each pot has to be mentioned.
- Why 0.6 g of urea was used? Add references to prove this concentration is the best for the experiment
- It will be useful to add photos of the experimental plants before treatments and after the treatments including control plants.
- Lines 73-74 – reference is needed – why you used these concentrations.
- How many plants per one biological replicate was used? 4 leaves per plant , so you need many plants to assess each time point each parameter…
- Why you used 4-leaf plants? It has to be explained. Aren’t they too young for the experiment.
- Why the duration of the waterlogging was 21 days – has to be explained, add references.
- What air temperature and light intensity were during the experiment? Figures with these data have to be added.
- Soil in pots: What is the soil pH, and other soil characteristics definitely have to be added. Without these details your study can not be reproducible.
- How you perform waterlogging stress – add details.
- SPAD – the reference and equation of this index has to be provided.
- Lines 101 – 102 – methods of the analysis have to be provided in details. How the samples were prepared, how the elements were analyzed. Now the study doesn’t look reproducible.
RESULTS
- The figures have not presented the statistically significant differences. Only bars are presented on the figures. What is represented by the bars, standard error or deviation – also does not clear.
- Thus, the differences are does not supported statistically thus the conclusions are not justified.
Reviewer 4 Report
Comments and Suggestions for Authors
Review Plants 3582495
In a study submitted for publication in “Plants” the authors investigated the effectiveness of fertilizer supplementation on alleviating waterlogging damages in canola. To monitor the effects of nitrogen application in a timing strategic manner the authors examined stomatal conductance, SPAD and photochemical quantum yield of photosystem II. They came to a conclusion that nitrogen application before waterlogging significantly improved these physiological parameters They also revealed that applying nitrogen fertilizer before waterlogging stress support K, Mg and Ca levels but had no effect on reducing Mn and Fe accumulation. Hence, elemental toxicity, which is typically associated with soil waterlogging, was not a factor in nitrogen-induced waterlogging alleviation.
While it is true that rapeseed varieties may benefit from nitrogen fertilization prior to waterlogging I, at least in terms of photosynthesis rates and element accumulation, this fact alone is not sufficient to understand the mechanisms that help mitigate crop damage from waterlogging.
The authors should include data on quantitative trait loci associated with tolerance-related traits as well as on transcriptional and translational regulations of specific genes and signaling pathways associated with the role of nitrogen application in plant adaptation to waterlogging.
It is possible that the conducted studies are more in line with the profile of “Agronomy”, although it cannot be ruled out that publication of the manuscript in this journal requires additional experiments in field conditions.
Reviewer 5 Report
Comments and Suggestions for Authors
Unfortunately, the article does not give the appearance of a completed scientific study.
1) Line 79 missing table 1.
2) 2-3 points is very little for studying the dynamics of changes, because the title is "Optimising nitrogen supplementation for canola: timing strategies to mitigate waterlogging stress", it is necessary to introduce additional time points in the experiment.
3) Figures are difficult to perceive, the symbols are very small.
4) The discussion is speculative and not sufficiently supported by literary sources.
5) To assess the final impact of fertilization at different times (before and during waterlogging stress), it is necessary to identify the impact on plant productivity (final wet and dry biomass) or measure the yield.
Reviewer 6 Report
Comments and Suggestions for Authors
The submitted Manuscript is devoted to an important and interesting subject of effects of waterlogging on canola plants.
Unfortunately, the Manuscript suffers essentially from the part of methods etc.
- Table 1 is mentioned while it is absent:
in the middle of waterlogging, 7 days after waterlogging (MN) on day- 78 14; and 3) no additional fertilizer application (Base) (Table 1).
- Supplementary material is mentioned but not added.
- The abbreviations should be described in full from the very beginning, e.g. SPAD value meaning chlorophyll contents.
- The methods are too weakly described, temperature, intensity of light should be measured and described.
- The soil composition should be described in full.
- The only three figures are given when it’s expected more different figures for a good Journal with high impact factor of 4.0.
- Control measurements before waterlogging should be done and provided.
- The Reviewer would be happy to see the growth parameters of the plants, e.g. their height, weight etc.
- Control measurements without waterlogging have to be provided to compare the two plant varieties.
- The Reviewer expects more information for the good paper in Journal Plants.
- Figures have to be transformed, “applied fertilizer” to be given instead of “apply fertilizer”.
- More data points are to be measured, anatomy and structure of plants to be studied; basically the number of points is low to make significant conclusions.
- show marginally higher stomatal conductance than Base or MN plants (Figures 1c and d, 126 Table S2). 127
Table S2 is absent.
- Elemental toxicity to be proved.
- Absolute values to be shown, not %; more information to be provided.
- Suggest reject-resubmit adding extra information.
Reviewer 7 Report
Comments and Suggestions for Authors
The article explores the critical role of optimized nitrogen supplementation in mitigating waterlogging stress in canola, a key rotation crop in Australia's high rainfall zones. By examining fertilization timing, the study demonstrates that pre-waterlogging nitrogen application significantly enhances physiological responses, such as photosynthetic efficiency and macronutrient accumulation. These findings offer valuable insights into developing strategies to improve canola resilience in waterlogged conditions, addressing a vital challenge in sustainable crop management. However, the following issues are to be addressed.
Line 19: Your study focuses more on changes in element content rather than physiological responses.
Line 20: Nothing is mentioned about the methodology—whether it is a pot experiment or field experiment.
Line 21: The duration of waterlogging should be mentioned in the abstract.
Line 23: What does ‘support’ mean? Please revise it for greater clarity.
Line 24: Nothing is initially mentioned about ‘both cultivars.’ What are those cultivars? Are they both sensitive to waterlogging, or do they show divergence in waterlogging tolerance?
Line 27: Elemental toxicity should be better explained; otherwise, it might mislead readers.
Introduction:
Line 38: Mention the scientific name of canola in parentheses at its first occurrence.
Line 40: It is important to further explain the extent of sensitivity. Is there divergence among genotypes according to the literature? What about the effect of waterlogging duration (different extents of waterlogging) on canola growth, development, and yield? Shed more light on these aspects.
Line 45, 56: Define or use the full form (e.g., SPAD, MDA) at their first occurrence.
Line 49: What about the effects of other nitrogen forms (fertilizers)?
Importantly,elemental toxicity should be better explained. Consult and cite the following article: Doi: 10.1016/j.ncrops.2024.100034
M&M:
Line 68-69: Mention whether they have any difference in waterlogging tolerance or any known stress tolerance characteristics. Are they equally sensitive to waterlogging?
Line 72: Provide details of weather parameters, such as temperature, light duration and intensity (PPFD), humidity, precipitation, etc., during this period.
Line 80: What is the basis for selecting this dose? Also, why was urea chosen as the nitrogen source?
Figure 1: ‘Apply fertilizer’—perhaps you meant “Application of fertilizer.” The authors should determine statistically significant differences, mentioning P-values, and indicate them with letters or asterisks in all figures.
Results:
What are the effects on plant growth and biomass? These data are missing! Add them. Additionally, phenotype pictures under different treatments should be included. Consider adding correlations between parameters studied and nutrients. Also, PCA could provide better insights.
Discussion:
Divergences in responses of the two cultivars to waterlogging have not been adequately addressed.
Line 261-283: Consult and cite the following articles and shed light on N-Fe balance, crop yield, and nitrogen use efficiency here and in the introduction section: Doi: 10.1016/j.ncrops.2024.100047
As mentioned earlier, your study focuses more on changes in element content rather than physiological responses. Therefore, speculation on physiological aspects should be optimized, and the focus should be on nutrient homeostasis.